# Evidence for Electron Transfer from the Bidirectional Hydrogenase to the Photosynthetic Complex I (NDH-1) in the Cyanobacterium *Synechocystis* sp. PCC 6803

**DOI:** 10.3390/microorganisms10081617

**Published:** 2022-08-10

**Authors:** Jens Appel, Sean Craig, Marius Theune, Vanessa Hüren, Sven Künzel, Björn Forberich, Samantha Bryan, Kirstin Gutekunst

**Affiliations:** 1Department of Molecular Plant Physiology, Bioenergetics in Photoautotrophs, University of Kassel, D-34132 Kassel, Germany; 2Department of Biology, Botanical Institute, Christian-Albrechts-University, D-24118 Kiel, Germany; 3BBSRC/EPSRC Synthetic Biology Research Centre, The Biodiscovery Institute, University of Nottingham, Nottingham NG7 2RD, UK; 4Max-Plank Institute for Evolutionary Biology, August-Thienemann-Straße 2, D-24306 Plön, Germany

**Keywords:** hydrogen, photosynthesis, electron transport, photosystem I, respiration

## Abstract

The cyanobacterial bidirectional [NiFe]-hydrogenase is a pentameric enzyme. Apart from the small and large hydrogenase subunits (HoxYH) it contains a diaphorase module (HoxEFU) that interacts with NAD(P)^+^ and ferredoxin. HoxEFU shows strong similarity to the outermost subunits (NuoEFG) of canonical respiratory complexes I. Photosynthetic complex I (NDH-1) lacks these three subunits. This led to the idea that HoxEFU might interact with NDH-1 instead. HoxEFUYH utilizes excited electrons from PSI for photohydrogen production and it catalyzes the reverse reaction and feeds electrons into the photosynthetic electron transport. We analyzed hydrogenase activity, photohydrogen evolution and hydrogen uptake, the respiration and photosynthetic electron transport of Δ*hoxEFUYH*, and a knock-out strain with dysfunctional NDH-1 (Δ*ndhD1*/Δ*ndhD2*) of the cyanobacterium *Synechocystis* sp. PCC 6803. Photohydrogen production was prolonged in Δ*ndhD1*/Δ*ndhD2* due to diminished hydrogen uptake. Electrons from hydrogen oxidation must follow a different route into the photosynthetic electron transport in this mutant compared to wild type cells. Furthermore, respiration was reduced in Δ*hoxEFUYH* and the Δ*ndhD1*/Δ*ndhD2* localization of the hydrogenase to the membrane was impaired. These data indicate that electron transfer from the hydrogenase to the NDH-1 complex is either direct, by the binding of the hydrogenase to the complex, or indirect, via an additional mediator.

## 1. Introduction

All respiratory complexes I that catalyze the oxidation of NADH and reduction of a quinone share 14 core subunits [1]. The first structural study on the cyanobacterial complex, now termed photosynthetic complex I, demonstrated that it receives electrons directly from ferredoxin [2]. In contrast to the canonical respiratory complexes, the photosynthetic complexes of cyanobacterial and plant chloroplasts do not contain the three outer subunits known to bind and oxidize NADH [2,3]. These subunits contain five additional FeS clusters that are absent from the photosynthetic structure. Instead, this kind of beheaded complex harbors only the three innermost FeS clusters with a binding site for ferredoxin on the top made up of three small additional subunits (NdhO, NdhV, and NdhS) only found in cyanobacteria and plant chloroplasts [3,4].

The absence of the outer subunits was already noted some time ago and the hypothesis was put forward that three of the subunits of the pentameric cyanobacterial bidirectional hydrogenase (HoxEFU) might take over. It was suggested that the hydrogenase forms a large complex with the photosynthetic complex I [5,6]. In a number of other bacteria and archaea large hydrogenase complexes with strong homologies to the respiratory complex I exist [7,8].

Hydrogenases catalyze the formation of H_2_ from two electrons and two protons or the reverse reaction. Due to its low redox potential H_2_ is a versatile electron donor that can be used to reduce NAD(P)^+^ or different quinones. In the reverse reaction protons serve as electron acceptors, e.g., under fermentative conditions to oxidize ferredoxin and form H_2_. Since hydrogenases serve an important role in energy metabolism there is a plethora of different classes and subtypes in the microbial world differentiated based on their reaction partners [9]. Many microbial communities rely on the exchange of hydrogen and even though the composition of Earth’s atmosphere changed dramatically and hydrogen is now a trace gas with 0.5 ppm, there is widespread use of hydrogenases in bacteria for sustained energy supply from the air [10].

Cyanobacteria are known to harbor two different types of [NiFe]-hydrogenases; the uptake, and the bidirectional hydrogenase. While the uptake hydrogenase is expressed in diazotrophic strains to regain the reducing power of the H_2_ produced as a side product by the nitrogenase, the bidirectional enzyme is there to remove a surplus of electrons either during fermentation or when the cells restart photosynthesis from dark anaerobic conditions [11,12,13].

The cyanobacterial bidirectional hydrogenase accepts electrons from ferredoxin and flavodoxin while producing photohydrogen [14]. This is supported by the lower redox potentials of ferredoxin/flavodoxin compared to NAD(P)H that prohibit H_2_ production from the pyridine nucleotides at their normal physiological NAD(P)H/NAD(P)^+^ ratio. Another study showed that the diaphorase module of this hydrogenase accepts electrons from ferredoxins to predominantly reduce NAD^+^ to NADH; the K_M_ of NADPH was found to be about 25 times higher [15]. This is supported by previous investigations that also showed about six times higher rates of H_2_ production with NADH compared to NADPH [16]. Thus, this type of hydrogenase is a central metabolic hub that mediates electron transfer between the three redox pools ferredoxin/flavodoxin, H_2_ and predominantly NADH. In particular, NADH formation from reduced ferredoxin is peculiar for this enzyme and might open an additional channel aside the FNR or the transhydrogenases when the cells become strongly reduced. 

It has long been known that this type of hydrogenase is responsible for hydrogen evolution in the light (so-called photoH_2_) that is produced when dark anaerobic cells are suddenly exposed to strong light [17]. Recently, we showed that the enzyme is also working as an electron valve when the cells are shifted to high light intensities under aerobic conditions [18]. It remains unclear how the enzyme performs this task but NADH formation from reduced ferredoxin and its subsequent oxidation in respiration are a possible route.

In cyanobacteria not only one type of photosynthetic complex I exists. There are at least four different types of NDH-1 complexes in *Synechocystis* that contain different NdhD and NdhF subunits. Two of these complexes contain either NdhD1/NdhF1 or NdhD2/NdhF1 and are known to be involved in cyclic electron transport and respiration [19], while two additional complexes are known to be part of the carbon-concentrating mechanism (CCM, [20]) and contain either NdhD3/NdhF3 or NdhD4/NdhF4. Here we will name the complexes according to their NdhD subunits as NDH-1_1_, NDH-1_2_, NDH-1_3_ and NDH-1_4_.

Localization studies using GFP-labelled bidirectional hydrogenase found variable proportions of the enzyme in the cytoplasm and at the membrane [21]. No evidence for a direct interaction with the NDH-1 complex could be found in this study, although the hydrogenase was found to be clustered in so-called puncta at the membrane, which is reminiscent of localization studies of the cyanobacterial NDH-1 that found highly concentrated patches of the complex [22]. Thus, although the bidirectional hydrogenase is a soluble protein complex, its membrane attachment remains unclear and an interaction with NDH-1 is still an option.

This leaves us with two main unsolved issues for the cyanobacterial case. The first concerns the carbohydrate breakdown known for cyanobacteria. Except a possible route via the pyruvate:ferredoxin oxidoreductase [23], all others produce NAD(P)H but not reduced ferredoxin. In principle there are ferredoxins with higher redox potentials in the cyanobacterial cell that could be reduced by NAD(P)H, but it is not known if the FNR or any other enzyme could catalyze the required reactions and if the NDH-1 is able to oxidize them in return. So how do cyanobacteria submit electrons from glucose breakdown to their respiratory electron transport chain?

The second issue concerns the oxidation of hydrogen (also known as hydrogen uptake) that is catalyzed by the bidirectional hydrogenase after a short phase of photohydrogen evolution or when hydrogen is provided under anaerobic conditions in the light and photosystem II is blocked. Under these conditions, H_2_ serves as a reductant for CO_2_-fixation [11,12]. It still remains to be solved if the electrons that are delivered to the Calvin–Benson–Bassham (CBB) cycle are mediated either via ferredoxin (Fd) or NAD(P)H (H_2_- > Fd- > PQ- > PSI- > CBB (Figure 1C) or H_2_- > NAD(P)H- > CBB (Figure 1B)), or if they are fed into the PQ pool by a hydrogenase bound to the NDH-1 complex (H_2_- > PQ- > PSI- > CBB, Figure 1D).

In this study, we undertook a number of investigations of the hydrogenase activity, hydrogen evolution in the light, hydrogen uptake, electron transport, and respiratory activity of the deletion strains of *Synechocystis* sp. PCC 6803 that are impaired in different NDH-1 complexes or the hydrogenase. The results indicate that the absence of the bidirectional hydrogenase causes a down-shift in respiratory activity and that the absence of specific NDH-1 complexes impairs hydrogen uptake. The colocalization of a GFP-labelled HoxF and a YFP-labelled NdhM show that the hydrogenase is found less in the membrane regions in a mutant without NDH-1_1_ and NDH-1_2_. Although we present physiological evidence only, this study provides strong evidence for electron transfer from the bidirectional NiFe-hydrogenase to the NDH-1 complex.

## 2. Materials and Methods

### 2.1. Growth Conditions

All strains were inoculated in 50 mL BG-11 and placed in 100 mL Erlenmeyer flasks on a rotary shaker at 28 °C, 50 µE m^−2^ s^−1^ and 100 rpm. After several days of growth, the cells were inoculated into 200 mL BG-11 at an OD_750_ of 0.05 and placed into glass tubes bubbled with air at 120 µE m^−2^ s^−1^ at 28 °C. After four days of growth, cells were harvested by centrifuging at 4000× *g* for 5 min and then re-suspended in fresh BG-11 for measurement.

If not indicated otherwise we used OD_750_ as a reference in all the measurements as it is a good proxy for cell dry weight [24] and because we did not find large differences between wild type and mutant strains concerning their chlorophyll content.

### 2.2. Construction of Mutants

Constructs were assembled in pBluescript SK- (Addgene, Watertown, MA, USA). The vector was cut with SacI and KpnI (Thermofisher Scientifc, Dreieich, Germany) and the PCR products including the resistance cassette were introduced by Gibson assembly [25] in a single step. All the primers used in this study are shown in Appendix A. After sequencing, the respective constructs were transformed into glucose-tolerant *Synechocystis* sp. PCC 6803 wild type cells as described [26]. After re-streaking the resulting transformants at least four times, the clones were tested by PCR or Southern hybridization for complete segregation (Appendix A).

In this study we used a deletion strain without any of the three type 2 dehydrogenase genes (*ndbA* (*slr0851*), *ndbB* (*slr1743*), and *ndbC* (*sll1484*)) named Δ*ndh-2*, and a strain lacking only *ndbA* and *ndbC*, Δ*ndbA*Δ*ndbC*. Since there is evidence that NdbB is only used as an oxidoreductase to reduce prenyl naphthoquinones and prenyl benzoquinones during the biosynthesis of phylloquinone and plastoquinone and the respective deletion mutant Δ*ndbB* is lacking phylloquinone [27] we investigated parallel to the complete knock-out strain (Δ*ndh-2*) also the one still containing *ndbB* to make sure that the lack of phylloquinone biosynthesis does not result in additional effects on electron transport and hydrogen metabolism. In general, both strains behaved very similarly and showed only small deviations that are shown in the different figures.

### 2.3. Oxygen Measurements

To measure oxygen consumption the cell density was adjusted to 20 µg chlorophyll/mL and 1.6 mL was placed in the cuvette of the Dual-KLAS/NIR (Walz, Effeltrich, Germany). An oxygen microsensor OX-50 (Unisense, Aarhus, Denmark) was placed in a lab-made holder that allowed for the insertion of the sensor tip from above into the solution while simultaneously concealing the surface of the solution from gas exchange with the surroundings. The data were acquired with a time resolution of 0.2 s and the stirrer was on all the time. To eliminate photorespiration, 5 mM NaHCO_3_ was added as well as 10 mM glucose when its effect on respiration or oxygen evolution was to be measured. Before measurements the oxygen sensor was calibrated as described by the manufacturer.

### 2.4. PhotoH_2_ Measurements

Hydrogen was measured using a microsensor H2-50 (Unisense, Aarhus, Denmark) and the same holder as described was used. Before starting the measurements, the sensor was calibrated as described by the manufacturer. The cell density was adjusted to 20 µg chlorophyll/mL and 1.6 mL was placed in a cuvette of the Dual-KLAS/NIR or the Multicolor-PAM (Walz, Effeltrich, Germany). The total volume also contained 40 U/mL glucose oxidase, 50 U/mL catalase, and 10 mM glucose to induce anaerobiosis. Cells were kept in darkness for 5 min to monitor fermentative H_2_ production and then a light with 750 µE/m^2^/s was switched on. PhotoH_2_ production and the subsequent H_2_ oxidation were followed until completion.

### 2.5. Hydrogen Uptake

In this case 20 µM DCMU was added to the sample and it was kept anaerobic by including 10 mM glucose, 40 U/mL glucose oxidase, and 50 U/mL catalase. Before inserting the H_2_-sensor, H_2_-saturated BG-11 was used to dilute the sample to 20 µg chlorophyll/mL. Thereby H_2_ concentrations between 400 and 500 µM were reached. In a 5 min dark period H_2_ loss due to diffusion was followed and then a light intensity series from 16 to 420 µE m^−2^ s^−1^ was applied. Each intensity was measured for about 3 min and the resulting slope of H_2_ uptake was corrected by the diffusion determined in the dark. Except for a brief mixing period at the start of the experiment, the stirrer was switched off to prevent the high diffusive loss of H_2_.

### 2.6. Electron Flow Determination through Photosystem I

To determine electron flow through photosystem I we used dark-interval relaxation kinetics (DIRK) as described [28]. To this end the Dual-KLAS/NIR instrument was used. It was set to Fast Acquisition and after preincubation of the cells for 1 min at the respective light intensity to ensure a steady-state, DIRK measurements were started. During the following period the light was repeatedly shut off for 25 ms every 0.3 s for 400 times using a Fast-Kinetic Trigger File in combination with a Fast-Kinetic Multi-Run (4000 points, 0.5 ms resolution). The resulting traces were averaged to reduce noise. Based on the rates of re-reduction of P700 and plastocyanin in the dark intervals it is possible to calculate the electron throughput of PSI as µM e^−^/s or e^−^/PSI/s. For further details please refer to Reference [28].

### 2.7. State Change Measurements

Cell suspensions were diluted to 2.5 µg chlorophyll/mL and 1.25 mL of this suspension was placed in a stirred cuvette of the Multicolor-PAM (Walz, Effeltrich, Germany). The intensity of the measuring light and the gain were set to one. The actinic light and the measuring light were red with a maximum at 625 nm. After switching on the measuring light to determine F_0_ one saturation pulse of 3500 µE/m^2^/s was given. Subsequently, an actinic light of 110 µE/m^2^/s was switched on and after 1 min another saturation pulse was given (F_m−glc_). During a following dark phase of 30 s, glucose was added to 10 mM and after another 3 min of actinic light a second saturation pulse was applied (F_m+glc_). To acquire a measure for the state change we used the formula (F_m−glc_ − F_m+glc_)/F_0_.

### 2.8. Confocal Laser Scanning Microscopy and Colocalization

Cyanobacterial cultures were grown at 30 μmol photon m^−2^ s^−1^ in BG-11 at 150 rpm. Anoxia was achieved by supplementing BG11 with 10 mM glucose and 16 U mL^−1^ glucose oxidase and 20 U mL^−1^ of catalase. A 10 μL aliquot of each culture was spotted onto BG-11 agar, smeared into a patch of approximately 0.5 cm^2^, and left to dry. Agar cubes of approximately 1 cm^2^ were cut and placed on a 35 mm glass bottom dish (MatTek Life Sciences). All experiments were carried out in triplicate.

Cells were imaged using the ZEISS Elyra 7 (Carl Zeiss AG, Oberkochen, Germany) with a 63× water immersion objective lens, photosynthetic pigments were excited at 633 nm, and the fluorescence was measured between 695–720 nm. The wavelengths for GFP and YFP excitation were 488 nm and 514 nm, respectively, and fluorescence was measured between 498–517 for GFP and 525–544 nm for YFP.

Photosynthetic pigments found in both Photosystem I and Photosystem II contain chromophores, which result in background autofluorescence. GFP and YFP both have strong absorption and emission spectrums, which bleach easily under excessive laser illumination at 488 nm and 514 nm [29]. To determine the fluorescence, which directly results from both the GFP and the YFP, images were recorded (pre-bleach), and then the laser intensity was increased for ten seconds to bleach both the GFP and the YFP (post bleach). Post bleach images were then recorded for both the GFP and the YFP. The post bleach images were subtracted from the pre bleached images, providing a true depiction of the distribution of GFP and YFP in the cell (Appendix A) [29]. Appendix A depicts a wild type control (no GFP/YFP); images were captured, and then the cells were subject to increased illumination for 10 s. There was no impact on the background autofluorescence from the photosynthetic pigments and no contribution from these pigments to the GFP or YFP channels (Appendix A).

Images were processed using the Fiji software (version 2.4.0 available at https://imagej.net/software/fiji/). The wild type images were tabulated first, to establish the background fluorescence and set the thresholds for colocalization. The image channels were split; the post-bleached fluorescent protein images were subtracted from the pre-bleached images to generate a composite image. The contrast settings in the fluorescent protein channels were set to optimize visualization and the contrast settings were applied to all other images for that channel. Colocalization was quantified by assessing the fluorescence intensity of each pixel relative to the individual channels, using the colocalization function in Fiji. The Manders coefficient was used to determine the degrees of colocalization with the variance between conditions and strains being measured using the *t*-test, as previously described [30]. 

## 3. Results

### 3.1. Dark Respiration

The dark oxygen uptake of different deletion mutants of *Synechocystis* sp. PCC 6803 was measured (Figure 2). A mutant without any of the structural genes of the bidirectional hydrogenase (Δ*hoxEFUYH*) shows a lowered oxygen uptake. A similar low oxygen uptake is only visible in a strain without NDH-1_1_ and NDH-1_2_ (Δ*ndhD1*Δ*ndhD2*) and a strain without any of the terminal respiratory oxidases (Δ*ox*) while all other strains such as those without the type 2 NADH-dehydrogenase (Δ*ndh-2*, Δ*ndbA*Δ*ndbC*), and those lacking either NDH-1_1_ or NDH-1_2_, show an oxygen uptake close to wild type cells. The strain lacking only the large hydrogenase subunit (Δ*hoxH*) shows a reduced oxygen uptake but this reduction is not as strong as in Δ*hoxEFUYH* or Δ*ndhD1*Δ*ndhD2*. Surprisingly, the Δ*ndhD3*Δ*ndhD4* has a much higher respiratory oxygen consumption than all other strains.

The addition of glucose to cells in the dark induces an increase in oxygen consumption. This increase is also visible in Δ*hoxEFUYH* and Δ*hoxH* (Appendix A).

NDH-1_1_ and NDH-1_2_ are known to be the major NDH-1 complexes involved in respiration [19]. This is corroborated by our results since the Δ*ndhD1*Δ*ndhD2* shows an oxygen uptake similar to a mutant without terminal respiratory oxidases (Δ*ox*). We attribute the highly increased respiration in the Δ*ndhD3*Δ*ndhD4* to either a kind of compensatory upregulation of the respiratory complexes NDH-1_1_ and NDH-1_2_, or a need for higher respiration/ATP production due to the lack of the CCM. The reduction of respiration in the Δ*hoxEFUYH* in particular is surprising and indicates that the diaphorase (the NAD(P)H- and ferredoxin-handling part of the hydrogenase) also has a role in introducing electrons into the respiratory electron transport.

The reduction of oxygen uptake in the Δ*hoxH* is also interesting, and is probably due to a destabilization of the whole enzyme due to the lack of the large subunit. We attribute the increase in dark respiration upon the addition of glucose to an increased oxidative pentose phosphate pathway under these conditions. Consequently, NADPH production would increase and could be degraded via FNR- > ferredoxin- > NDH-1. This is a pathway that could be pursued in the dark when the PQ pool is not reduced by PSII since FNR is known to work in the reverse direction [31].

### 3.2. PhotoH_2_ Production

In Figure 3, original traces of measurements of photoH_2_ production of the different strains are shown as recorded with a H_2_ electrode. For a detailed analysis of the different phases of H_2_ production and uptake including the standard deviation see Appendix A. To induce photoH_2_ production, the cells were kept in darkness under anaerobic conditions for 5 min. After this dark phase the light was switched on with an intensity of about 700 µE/m^2^/s.

The two strains that lack *ndhD1* produce higher amounts of H_2_ in the light. While Δ*ndbA*Δ*ndbC* and Δ*ndhD2* are not different to wild type cells, the Δ*ndhD3ndhD4* shows diminished photoH_2_ production.

When comparing the different strains, it is important to note the Δ*ndhD1*Δ*ndhD2* double mutant consistently shows a long phase of photoH_2_ production and the lowest rates of H_2_ uptake (Figure 3 and Appendix A).

Direct measurements of the hydrogenase reveal that the activities of all the strains are in a small range around wild type activity (Appendix A). The Δ*ndhD1*Δ*ndhD2* mutant has about 80%, and the Δ*ndhD3*Δ*ndhD4* and Δ*ndbA*Δ*ndbC* both have about 70% of the activity of wild type cells.

### 3.3. Electron Flow through PSI during H_2_ Uptake

To follow H_2_ uptake while measuring electron flow through PSI, anaerobiosis was induced by glucose, glucose oxidase, and catalase, as demonstrated above, and DCMU was added to block PSII activity. Under these conditions H_2_ is the electron donor and photosynthesis including only PSI is running. H_2_ consumption was measured by a H_2_ electrode and electron flow was determined by DIRK (dark interval relaxation kinetics) measurements as described earlier [28].

Figure 4 on the one hand shows the increase of electron flow through PSI as calculated by subtracting the electron flow as measured without H_2_ from the electron flow with H_2_ (black symbols), and on the other hand the number of electrons generated by H_2_ oxidation (red symbols). Please note that a strain without hydrogenase does not show an increased electron flow through PSI after the addition of H_2_ [28].

While all the strains measured show a more or less parallel increase of PSI electron throughput compared to H_2_ oxidation, the Δ*ndhD1*Δ*ndhD2* double mutant increases its H_2_ uptake to only about 40% of the electron flow. To explain this deviating behavior, we investigate the electron flow of the different strains (Figure 4 and especially Appendix A) more closely. From the DIRK measurements made in parallel to the H_2_ uptake measurements it is possible to calculate the NADPH/ATP ratio formed. When the cells fix CO_2_ they have to produce a certain ratio of NADPH to ATP to fulfil the needs of the CBB cycle [32]. We found that the sum of linear and cyclic electron transport as determined by DIRK measurements when the cells perform oxygenic photosynthesis should yield a ratio of 2.2 [28]. This is very close to the value of 2.13 calculated on the basis of stoichiometric considerations for the biomass generated by *Synechocystis* [33]. Table 1 shows the NADPH/ATP ratio calculated from the data in Figure 4 and Appendix A. In this case we assumed that the major entry site for electrons into the electron transfer to PSI is the NDH-1 complex transferring a total of four protons per electron. Only in case of the Δ*ndhD1*Δ*ndhD2* mutant the entry site must be the type 2 dehydrogenases (NDH-2) that are not able to couple electron transfer to proton translocation. Thus, in this case, the transfer of two protons per electron was considered. These results confirm that the NDH-1 complex is not used in the Δ*ndhD1*Δ*ndhD2* but in all others.

The calculated values are in a range from about 2.6 to 4.2 and probably differ from one mutant to the other because of the deviating use of different electron transport pathways and/or the inhibition of specific pathways due to the specific mutation. Under the conditions used here (pH 8 and 5 mM HCO_3_**^−^**) the Δ*ndhD1*Δ*ndhD2* and the Δ*ndhD3*Δ*ndhD4d* in particular—because of their impaired NDH-1 complexes—might be forced to use a higher proportion of ATP for the import of bicarbonate and thus need a higher ATP/NADPH ratio.

Based on these calculations the Δ*ndhD1*Δ*ndhD2* strain would generate an ATP/NADPH ratio of eight according to our measurements when using the NDH-1 complex. This is far outside the range of the other strains and corroborates that this strain, in contrast to the others, is not using this complex for electron transfer when using H_2_ as electron donor for CO_2_ fixation.

When the cells take up H_2_ under these conditions it is consumed completely. From the concentration curves measured against time it is possible to compute the rates of H_2_ uptake for every concentration. We plotted the rates against the respective concentration (see Appendix A for an example of such a plot). H_2_ consumption by the bidirectional hydrogenase shows a cooperative behavior and when fitted with a Hill equation we acquired a K_M_ of 11.5 µM ± 1.5 and 9.0 µM ± 1.8 for wild type cells and the Δ*ndhD1*Δ*ndhD2* double mutant, respectively. While these two values are very close to each other and inside the experimental error range, the V_max_ value of 6.9 nmol H_2_/L/min/OD_750_ ± 0.7 for wild type cells is more than twice as large as 2.9 nmol H_2_/L/min/OD_750_ ± 0.2 for the mutant. These results indicate that the hydrogenase itself shows the same affinity for H_2_ in both strains but that the electron transfer to a possible acceptor is impaired or different in the mutant strain.

Taken together these results suggest that NDH-1_1_ and NDH-1_2_ specifically are needed for H_2_ uptake. Since it is well known that NAD^+^ is a good electron acceptor for electrons from the hydrogenase [11,14,16] its reduction to NADH and subsequent oxidation by the NDH-2 might be the alternative pathway in the Δ*ndhD1*Δ*ndhD2* mutant. It seems highly improbable that ferredoxin 1 (Ssl0020*)*, which is an electron donor to the hydrogenase [14], could be a mediator between the hydrogenase and the NDH-1 complex for H_2_ uptake since it should be reduced to a certain extent by PSI under photosynthetic conditions. Using the Nernst equation, it is possible to calculate, which concentration of H_2_ is in equilibrium with a certain ratio of reduced to oxidized ferredoxin (Fd_red_/Fd_ox_). If we assume that this ratio is in the range from 1.5 to 0.5 and if we take into account that the intracellular pH is about 7.3 [34], concentrations between 360 and 42 µM could be reached at the lowest, and even at a ratio of 0.1 (which seems unlikely during photosynthesis/CO_2_-fixation) the H_2_ concentration would still be 1.7 µM (see Appendix A). Since the H_2_ level drops to about 0 µM during uptake (Figure 3) ferredoxin 1 is an unlikely electron mediator. Instead, it is much more likely that NAD^+^ is the electron acceptor in case of the Δ*ndhD1*Δ*ndhD2* strain and PQ via the NDH-1 complex in all other strains. This is supported by the normal H_2_ uptake activity of the NDH-2 deletion strain, which is known to accept electrons from NADH [35] and that should not be possible if all strains were using NAD^+^ as an electron acceptor. 

The main conclusion we draw from these results is that the electron transfer from the hydrogenase to the NDH-1 during H_2_ uptake is either direct and results in PQ reduction, or mediated by a carrier that is neither ferredoxin 1 nor NADH in wild type cells.

### 3.4. Glucose Utilization in the Light

If the hydrogenase would directly couple to the NDH-1 complex as suggested by the above measurements, its diaphorase should also funnel electrons into the intersystem chain. This is supported by our dark respiration measurements (Figure 2).

Here we made measurements of oxygen consumption in the light, without and after the addition of glucose, and of the electron flow through PSI. As shown in Figure 5, wild type cells and a number of deletion strains, including Δ*ox* without respiratory terminal oxidases; Δ*ndh-2* without the type 2 dehydrogenases; Δ*ndhD1*, without NDH-1_1_; Δ*ndhD2* without NDH-1_2_; Δ*ndhD1*Δ*ndhD2* without NDH-1_1_ and NDH-1_2_; Δ*ndhD3*Δ*ndhD4* without NDH-1_3_ and NDH-1_4_; Δ*hoxEFUYH* without any of the hydrogenase subunits; and Δ*hoxH* without the large hydrogenase subunit, were measured at different light intensities. In the experiment the cells were exposed to a specific light intensity for 1 min to reach a steady state. In the approximately 2 min following, changes in the rate of oxygen turnover were recorded and the electron transfer through PSI was determined by DIRK measurements. Subsequently, 10 mM glucose was added and after another 1.5 min under continuous light the oxygen and DIRK measurements were repeated.

After the addition of glucose, wild type cells decrease their oxygen evolution. Upon the addition of glucose the cells receive a surplus of carbon skeletons and reducing equivalents. To balance the consumption and further conversion of these metabolites the cells have three possibilities. They could either (1) increase respiration, (2) turndown the activity of PSII, (3) increase the consumption of low potential electrons by other acceptors, or undergo a combination of these options. In all three cases oxygen evolution should decrease in the presence of glucose. As is visible in Figure 5 increased respiration (option 1) cannot be the reason since the mutant of the respiratory terminal oxidases (Δ*ox*) and the one without the type 2 dehydrogenases (Δ*ndh-2*) show the same suppression of oxygen evolution as the wild type cells.

Concerning the bidirectional hydrogenase, it is striking that none of the hox mutants (Δ*hoxEFUYH* and Δ*hoxH*) show a decreased oxygen evolution. The most straight forward explanation is that under these conditions the diaphorase delivers electrons from glucose breakdown into the intersystem chain in wild type cells but not in the *hox* deletion strains. These electrons should be at the level of NADH or NADPH. A reduction of ferredoxin 1 (by HoxEFUYH) and its subsequent oxidation by the NDH-1 complex seems to be highly unlikely because of the more positive redox potential of the pyridine dinucleotides. Thus, a direct coupling of the diaphorase to the NDH-1 is much more probable.

All the mutants of NdhD subunits tested here also do not show the suppression of oxygen evolution, supporting the idea that the NDH-1 complex is involved in the oxidation of the additional reducing equivalents. The Δ*ndhD3*Δ*ndhD4* double mutant shows an increased oxygen consumption in the dark and at low light intensities. Therefore, the NDH-1_1_ and NDH-1_2_ complexes seem to deliver electrons into the intersystem chain especially in the dark and at low light, whereas the NDH-1_3_ and NDH-1_4_ complexes take over at higher light.

DIRK measurements (Appendix A) indicate that in all the strains showing the suppression of oxygen evolution (WT, Δ*ox*, Δ*ndbA*Δ*ndbC*) the electron transfer through PSI is slightly increased while in all other strains it is either the same or decreases.

If the breakdown of glucose results in a higher number of electrons introduced into the intersystem chain via the NDH-1 complex, we can expect that the plastoquinone pool might become more strongly reduced. Since the redox state of the PQ pool controls excitation energy distribution between the two photosystems [36] the surplus of electrons might induce state change. More specifically, we would expect that excitation energy/antenna are shifted away from PSII to decrease its activity (option 2).

### 3.5. State Change Due to Glucose Addition

Parallel PAM-fluorescence measurements were performed for all the strains without and with glucose to monitor their putative state changes. By giving saturation pulses of high light, the excitation of PSII before and after glucose was measured. As visible in Figure 6, all the strains that show the increased oxygen consumption after the addition of glucose also show a state change and remove antenna from PSII (the F_m_ without glucose is higher), while all those that do not show this effect have a reduced state change or even have a negative value (Δ*ndhD1*Δ*ndhD2*) that indicates that more excitation energy is funneled into PSII. 

Taken together, these results indicate that glucose addition results in a stronger reduction of plastoquinone and suggest that wild type cells decrease the amount of excitation energy that is directed to PSII. This will result in a lowered oxygen evolution. Even the mutant without any of the respiratory terminal oxidases shows the same state change behavior, showing that the oxidases are not important in the withdrawal of surplus electrons as previously shown in Figure 5. However, since net oxygen evolution is strongly diminished in this strain upon glucose addition, it is likely that other players such as the flavodiiron proteins are also involved and increase oxygen consumption through a Mehler-like reaction (option 3).

The negative state change seen in the Δ*ndhD1*Δ*ndhD2* is probably due to the higher amount of reducing equivalents generated by glucose breakdown that must be balanced by a higher ATP production. Since this strain is only able to increase ATP synthesis by increasing its linear electron transport driving the water–water cycle [28,37] it must shift a higher proportion of excitation energy to PSII.

Since state change is reduced especially in the Δ*hoxEFUYH* mutant this again indicates that electron flow into the PQ pool is impaired especially in the hox mutant without the diaphorase.

### 3.6. Effect of Glucose Addition on Electron Transport When PSII Is Inhibited by DCMU

Another possibility to measure the effect of glucose addition is to determine the electron flow in the absence and presence of glucose through PSI when PSII is blocked by DCMU. Under these conditions the PQ pool is strongly oxidized. This is obvious when comparing the oxidation level of PSI in the absence and presence of DCMU (Appendix A). In Figure 7 the results for all the deletion strains investigated here, including a strain lacking the succinate dehydrogenase (Δ*sdhB1*Δ*sdhB2*), are shown.

Previously we already showed that the Δ*ndhD1*Δ*ndhD2* mutant is unable to increase its electron flow upon glucose addition [23]. From Figure 7 it is obvious that all other deletion strains are able to increase their electron flow through PSI. Thus, specifically the complexes NDH-1_1_ and NDH-1_2_ are needed to re-oxidize the reducing equivalents and reduce PQ. It is not sufficient to delete one of these two complexes to block the electron flow triggered by glucose addition. This is different to normal photosynthetic conditions (Figure 5) where Δ*ndhD1* and Δ*ndhD2* already show reduced oxygen evolution which indicates that both are required to introduce the electrons from glucose breakdown.

Because the PQ pool is strongly oxidized under these conditions and the electron flow through PSI is much smaller than when PSII is active [28], ferredoxin 1 could be an electron donor to the NDH-1 complexes. This would also explain why the Δ*hoxEFUYH* mutant is able to increase its electron flow after the addition of glucose since the FNR could replace the diaphorase and work in reverse, as previously discussed concerning the respiration measurements.

### 3.7. Colocalization of HoxF and NdhM

To determine the colocalization of the bidirectional hydrogenase and NDH-1 in vivo, we created C terminal gene fusions between HoxF and Green Fluorescent Protein (GFP), and NdhM and Yellow Fluorescent Protein (YFP), with the HoxF protein being expressed at the native chromosomal *hoxF* locus and NdhM being expressed at the native *ndhM* locus. As previously shown [21], HoxF–GFP was still fully functional being incorporated into fully assembled complexes and H_2_ production was still comparable to the wild type strain. NdhM–YFP was also fully functional as demonstrated [22]. Confocal laser scanning microscopy (CLSM) was utilized to determine the localization of both HoxF–GFP and NdhM–YFP relative to the thylakoid membrane, identified from chlorophyll fluorescence. The resolution of CLSM is not sufficient to localize complexes to the nm scale and to judge if both are in direct vicinity. However, it is possible to acquire a measure of the cellular distribution of proteins/complexes and subsequently calculate the match of two different populations of complexes. To this end, multiple images were acquired per condition and the post bleach image was subtracted from the pre bleach image; images were analyzed in Image J utilizing the colocalization function to calculate the Manders coefficient. This value gives a measure of the congruence of the two distributions [38,39].

Under photoautotrophic growth at 30 μmol photon m^−2^ s^−^^1^, Hox colocalized to the thylakoid membrane as identified from the chlorophyll fluorescence (Appendix A), with a Manders coefficient (M) of 0.7107 (Figure 8). Hox was present in two populations; a dispersed population, and puncta, correlating with the distribution already seen [21] (Appendix A). Following the addition of glucose, glucose oxidase and catalase to induce anoxia, Hox showed a higher degree of colocalization with the photosynthetic pigments (M = 0.9989) (Appendix A), indicating a change in the distribution of Hox. The increase in Hox and photosynthetic pigment colocalization under anoxia compared to photoautotrophic growth was significantly enhanced (*p* = 0.09). NDH-1 was shown to be colocalized with the photosynthetic pigments under photoautotrophic growth (M = 0.7099) (Appendix A). Following anoxia, the colocalization of NDH-1 and chlorophyll increased (M = 0.9979), which was similar to the redistribution seen in Hox (Appendix A). NDH-1 colocalization with the photosynthetic pigments was significantly increase upon anoxia (*p* = 0.018). Both Hox and NDH-1 were colocalized at the thylakoid membrane (M = 0.7782) under photoautotrophic growth (Appendix A). This association increased following anoxia (M = 0.9212), however both Hox and NDH-1 still maintained two distinct sub populations (Appendix A). Overall, anoxia significantly increased the colocalization of Hox and NDH-1 (*p* = 0.002).

The significantly lower level of colocalization between Hox and NDH-1 upon anoxia in Δ*ndhD1*Δ*ndhD2* when compared to WT (*p* = 0.0001) suggests the loss of a functional interaction between NDH-1_1_ and NDH-1_2_ and the hydrogenase.

## 4. Discussion

Hydrogenases are extremely versatile redox enzymes mediating electron transfer between a large number of different reaction partners. The cyanobacterial bidirectional hydrogenase is an excellent example since it mediates electron transfer between hydrogen, ferredoxin/flavodoxin and NAD(P)^+^ [11,14,15,16]. It has long been known that this enzyme is also responsible for H_2_ uptake under anaerobic conditions when PSII is blocked/not working or at low light intensities [13,40,41]. In principle the electrons from H_2_ oxidation could follow three different routes. The first could involve the hydrogenase reducing ferredoxin, which would be re-oxidized by the NDH-1 complex; in the second, it could reduce NAD^+^ that in turn should be re-oxidized by the type 2 dehydrogenases since these are the only known to oxidize NADH in *Synechocystis* [35]; and in the third either electrons are directly delivered to the NDH-1 complex by the hydrogenase or they are passed on to the complex by another unknown mediator.

Our experiments indicate that H_2_ oxidation is closely linked to functional NDH-1_1_ and NDH-1_2_ complexes (Figure 4). The absence of these two complexes results in hydrogen uptake with the same apparent K_M_ but a significantly lower V_max_. Since the Δ*ndhD1*Δ*ndhD2* has a similar amount of hydrogenase (Appendix A) this indicates that it uses a different electron acceptor compared to wild type cells and all the other strains such as the Δ*ndhD1* and Δ*ndhD2* single deletion mutants, the Δ*ndhD3*Δ*ndhD4*, and the strain without the type 2 dehydrogenase (Δ*ndh-2*). Calculations of the ATP/NADPH ratio produced by the different strains according to their electron flow (Figure 4 and Table 1) indicates that all of them except the Δ*ndhD1*Δ*ndhD2* double mutant must use their NDH-1 complex during H_2_ uptake. If they do not use this complex, the ratio will not fit the requirements of the CBB cycle. 

The fact that hydrogen concentration drops to zero during H_2_ uptake excludes ferredoxin 1 as electron mediator since its redox potential under physiological conditions would not allow complete consumption of hydrogen (Appendix A). Since in the Δ*ndhD1*Δ*ndhD2* electron transfer does not involve the NDH-1 complex, this strain must use an alternative route. Most plausibly the hydrogenase reduces NAD^+^ in this strain and the formed NADH in turn is oxidize by the type 2 dehydrogenases. The latter are known to have a low activity and were originally assigned a purely regulatory role [42]. However, recent investigations on a *Synechocystis* strain that expresses the soluble hydrogenase of a Knallgas bacterium, which is known to exclusively reduce NAD^+^ [43] already suggested that the cells must be able to oxidize NADH during H_2_ oxidation [44].

In summary, our H_2_ oxidation experiments indicate that plastoquinone receives the electrons via the NDH-1 complex in all the strains tested but in the Δ*ndhD1*Δ*ndhD2*, NAD^+^ is used instead.

We found that the addition of glucose to wild type cells performing normal photosynthesis results in diminished oxygen evolution (Figure 5). Under these photomixotrophic conditions, glucose breakdown results in a higher number of electrons entering the intersystem chain. On the one hand this is seen by the slightly increased number of electrons passing PSI (Appendix A) and on the other by the excitation energy redistribution away from PSII (Figure 5). Whether the lowered oxygen evolution is only due to decreased PSII activity or results from a combination of reduced water oxidation and increased oxygen consumption remains to be seen. Since the mutant lacking all respiratory terminal oxidases (Figure 5) shows normal wild type behavior, respiratory oxygen uptake is not involved. Thus, an increased water–water cycle would be the only other alternative for diminished net oxygen evolution.

It is not surprising that the mutant without the two NDH-1 complexes (Δ*ndhD1*Δ*ndhD2*) known to be part of respiratory electron transport [19] does not reduce oxygen evolution upon glucose addition (Figure 5). Since the entry of reducing equivalents into the electron transfer is impaired in this strain (also see the electron flow through PSI, Appendix A) the PQ pool would not become more reduced and would not force the strain to dimmish its PSII activity or to channel surplus electrons into oxygen reduction. However, it is surprising that the two single mutants Δ*ndhD1* and Δ*ndhD2* already show the same effect while glucose addition on the state change in the Δ*ndhD1* is stronger compared to the Δ*ndhD2*. On the other hand, both single mutants do increase their electron flow through PSI after glucose addition when PSII is blocked (Figure 7). The Δ*ndhD3*Δ*ndhD4* increases oxygen consumption at low light intensities but not at higher light (Figure 5). It performs state change although it does not increase electron flow significantly after glucose addition in the presence of DCMU (Figure 7). Taken together, these results indicate that there is still a lot to learn about the functioning of the NDH-1 complexes and that there is some unknown cooperation between NDH-1_1_ and NDH-1_2_ causing a kind of interdependence between both and also between the complexes NDH-1_3_ and NDH-1_4_ and the two aforementioned complexes.

Most unexpectedly the *hox* mutants also do not decrease oxygen evolution after glucose addition (Figure 5), they do not increase electron flow (Appendix A), and they do not redistribute the excitation energy as strongly as wild type cells (Figure 6). This indicates that electron transfer into the intersystem chain is impaired in these strains. Glucose breakdown yields reducing equivalents predominately at the level of the pyridine nucleotides (NAD(P)H). Under photomixotrophic conditions, glucose is fed mainly via the PGI shunt into the CCB cycle. This shunt does not yield any reducing equivalents. However, some glucose is channeled into the OPP and yields two NADPH for each molecule [45]. These investigations have been conducted under steady state conditions. Although our measurements depict the reaction of the cells upon sudden glucose addition, suppression of oxygen evolution also indicates that more reducing equivalents become available. Obviously, the cells need to oxidize the surplus NAD(P)H formed and the strains without a functional bidirectional hydrogenase (Δ*hoxEFUYH* and Δ*hoxH*) are unable to. In this case it must be the diaphorase module that is responsible for this activity, and could deliver electrons from NADH oxidation into the electron transport chain, but also from NADPH, albeit with a lower affinity. 

The fact that the Δ*hoxH* strain also behaves similarly to the mutant without any of the *hox* genes (Figure 5 and Figure 6) during dark respiration (Figure 2) indicates that the stability or assembly of the complex might be impaired in the absence of the hydrogenase large subunit. Especially in this case the redox potential of NAD(P)H does not allow for the reduction of ferredoxin and thus delivery from the hydrogenase to the NDH-1 complex must either be direct or conducted by a different mediator with a more positive redox potential (Figure 9).

We interpret a very recent study that found a higher NADPH level and higher contents of glycogen and polyhydroxybutyrate (PHB) in a Δ*hoxYH* deletion strain [46] in the same way as stated above. The interaction of the bidirectional hydrogenase including its diaphorase is probably impaired in this strain, resulting in an inhibition of the respiratory breakdown of all these metabolites. Under the conditions used in this study we never found measurable H_2_ levels thus excluding the idea that the hydrogenase is used to drain low potential electrons to H_2_ as stated by the authors.

In a previous investigation, it was shown that a population of the bidirectional hydrogenase is localized to the membrane while others are soluble inside the cytoplasm [21]. Membrane binding was attributed to the HoxE subunit although it is a soluble protein. Our colocalization studies suggest that attachment to the membrane is not direct but due to the interaction of the hydrogenase with the NDH-1 complex (Figure 8). The loss of HoxE might perturb this interaction, as the loss of HoxH also seems to impair the functioning of the whole enzyme (result in Figure 2, Figure 5 and Figure 6).

It is highly attractive to hypothesize that attachment of the hydrogenase to the NDH-1 complex is controlled by the redox conditions inside the cell and thereby the reactions it catalyzes. The changes seen in the colocalization of NDH-1 and photosynthetic pigments (Figure 9B) are interesting in this respect as well, as they might indicate that the complex might migrate closer to the photosystems in one condition and in others move in membrane areas with less photosystems. In case of the *ndhD1*Δ*ndhD2* one could hypothesize that the NDH-1_3_ and NDH-1_4_ complex are usually closer to the photosystems (Figure 9B) since they are involved in the CCM and needed during photosynthesis while the NDH-1_1_ and NDH-1_2_ complexes are also part of respiration and would not all be required close to the photosynthetic electron transport.

The scenario of NAD(P)H oxidation by the diaphorase of the hydrogenase as described above holds when PSII is active and also under dark conditions. In contrast to this, the inhibition of PSII in the light (Figure 7) or glucose addition in the dark (Appendix A) results in an increase in electron flow and increased respiration to wild type levels also in the *hox* mutants. We attribute this effect to an FNR working in the inverse direction since either the PQ pool is strongly oxidized in the light, the reductive power caused by rising NAD(P)H levels should increase, or both should occur. This effect could open an alternative pathway that would circumvent the diaphorase of the hydrogenase.

The photosynthetic complex I is part of a zoo of different transmembrane complexes that share a similar architecture in their membrane module (sometimes termed P-module). This module is used for proton translocation, which is driven by different redox reactions. It seems plausible that these complexes developed by increasing the redox potential difference these reactions span and thereby gradually increased the number of protons they can translocate [7,8]. One of the most primitive of these complexes might have oxidized reduced ferredoxin and produced H_2_ [47]. A step further in evolution reduced ferredoxin might have been oxidized using polysulfide (e.g., in *Pyrococcus* species) or F_420_ was oxidized by methanophenazine (as in some methanogens). When oxygen became available quinones could be reduced by NADH and the largest potential difference could be used by oxidizing ferredoxin by plastoquinone as it is performed by the photosynthetic complex I [7].

Hydrogen is an attractive electron donor and has about the same redox potential as ferredoxin 1. It seems consequential to harness its reducing power to drive plastoquinone reduction through a complex consisting of the photosynthetic complex I and the bidirectional hydrogenase. In addition, this complex, in combination with the hydrogenase, might have played an important role during the evolution of photosynthesis as an intermediary step for cells with a type I reaction center that used H_2_ as electron donor [48]. Today this is still performed by green sulfur bacteria, acidobacteria and heliobacteria [49]. Our results suggest that cyanobacteria are able to introduce H_2_ into the photosynthetic electron transport via a NDH-1 hydrogenase complex.

## Figures and Tables

**Figure 1 microorganisms-10-01617-f001:**
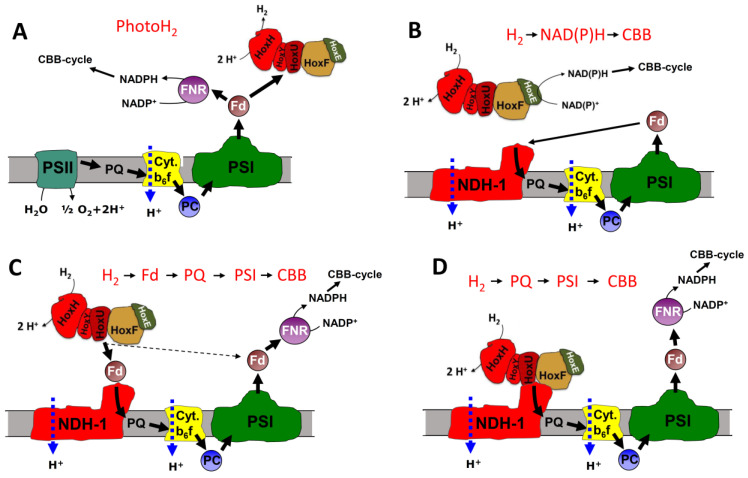
Schematic showing the different electron transport routes discussed. (**A**) Under anaerobic conditions when the CBB cycle is not activated the hydrogenase acts as an electron valve and produces photoH_2_ when photosynthesis starts from darkness. Because of PSII forming O_2_, conditions turn more oxidizing after a short time and H_2_ evolution turns into hydrogen uptake (for an example see Figure 3). The electrons gained from H_2_ oxidation could follow three different routes. (**B**) The hydrogenase could directly reduce NADPH, which would than serve the CBB cycle while cyclic electron transport provides the necessary ATP. (**C**) In an alternative pathway H_2_ oxidation could result in ferredoxin reduction that is then re-oxidized by the photosynthetic NDH-1 complex. (**D**) In the third option, the hydrogenase could be part of the NDH-1 complex and deliver the electrons directly to the PQ pool. Abbreviations: Fd = ferredoxin, FNR = ferredoxin NADP reductase, PSII and PSI = photosystem II and I, PQ = plastoquinone, Cyt b6f = cytochrome b6f complex, NDH-1 = photosynthetic complex I, CBB = Calvin–Benson–Bassham cycle.

**Figure 2 microorganisms-10-01617-f002:**
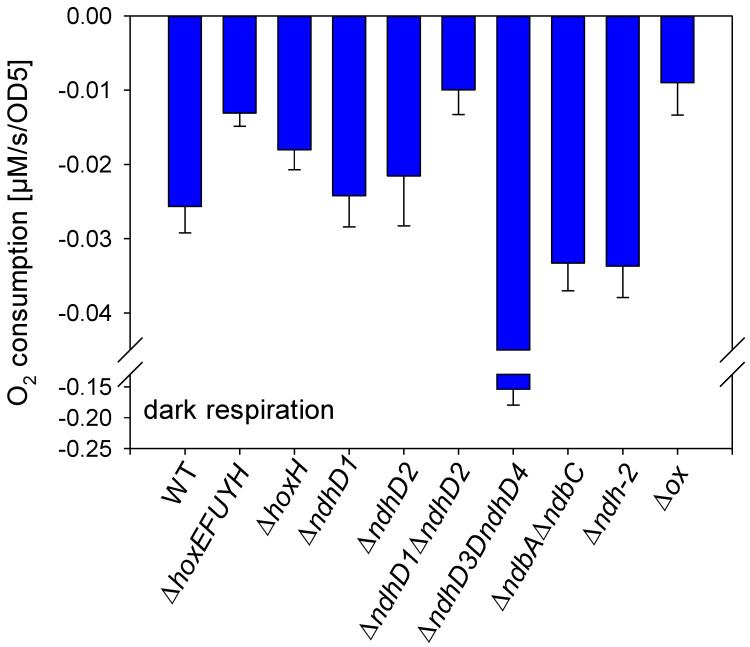
Dark respiration of different deletion strains of *Synechocystis* sp. PC 6803. Cell density was adjusted to the same OD_750_ and oxygen uptake measured for 5 min in the dark. The rate of oxygen uptake was taken from at least three different cultures of the same strain of the last 3 min. Error bars represent standard deviation.

**Figure 3 microorganisms-10-01617-f003:**
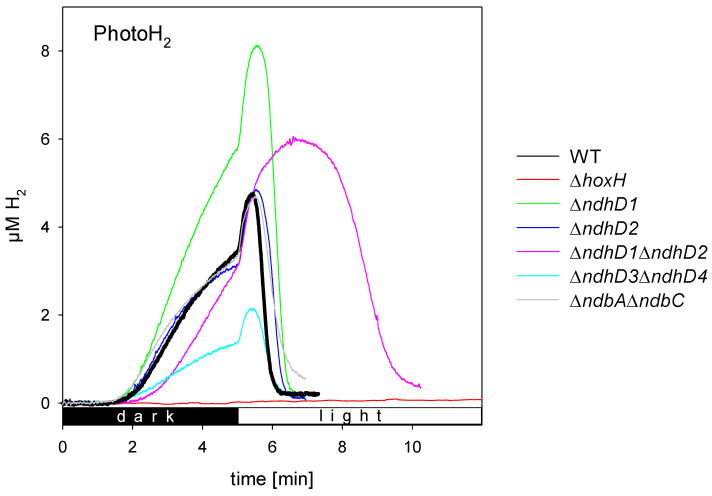
PhotoH_2_ production of different deletion strains of *Synechocystis* sp. PC 6803 in comparison to wild type cells and cells without a hydrogenase (Δ*hoxH*). Cell density was adjusted to the same OD_750_, anaerobiosis was induced by addition of 10 mM glucose, 40 U/mL glucose oxidase and 50 U/mL catalase. The original traces of wild type cells, Δ*hoxH* and the *ndhD* deletion strains are shown in comparison to a strain lacking the type 2 dehydrogenase (Δ*ndbA*Δ*ndbC*). For the variability between different cultures of the same strain please refer to Appendix A.

**Figure 4 microorganisms-10-01617-f004:**
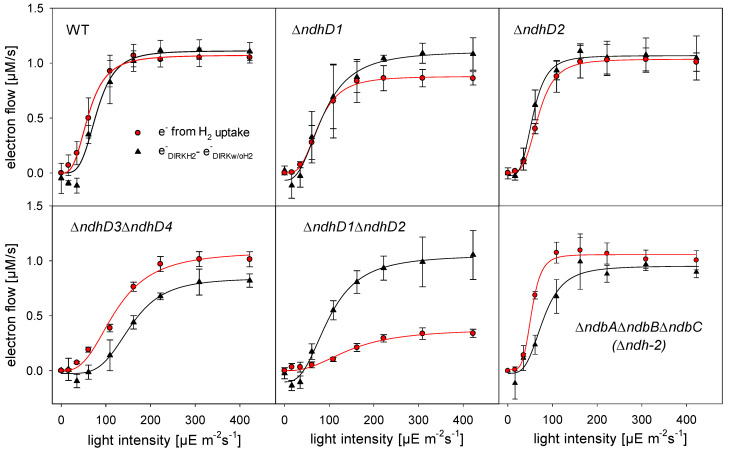
Parallel measurements of H_2_ consumption (by a hydrogenase electrode) and electron flow through PSI by DIRK (dark interval relaxation kinetics) measurements of different *ndhD* deletion strains and the deletion strain of the type 2 dehydrogenase (Δ*ndbA*Δ*ndbB*Δ*ndbC*). The black triangles and the black lines indicate the increase of electron flow through PSI due to the addition of H_2_ and the red circles and red lines indicate the number of electrons generated by H_2_ oxidation. The rates are given as µM electrons/s at a cell density of OD_750_ = 5.7. The error bars indicate the standard deviation of at least three independent cultures.

**Figure 5 microorganisms-10-01617-f005:**
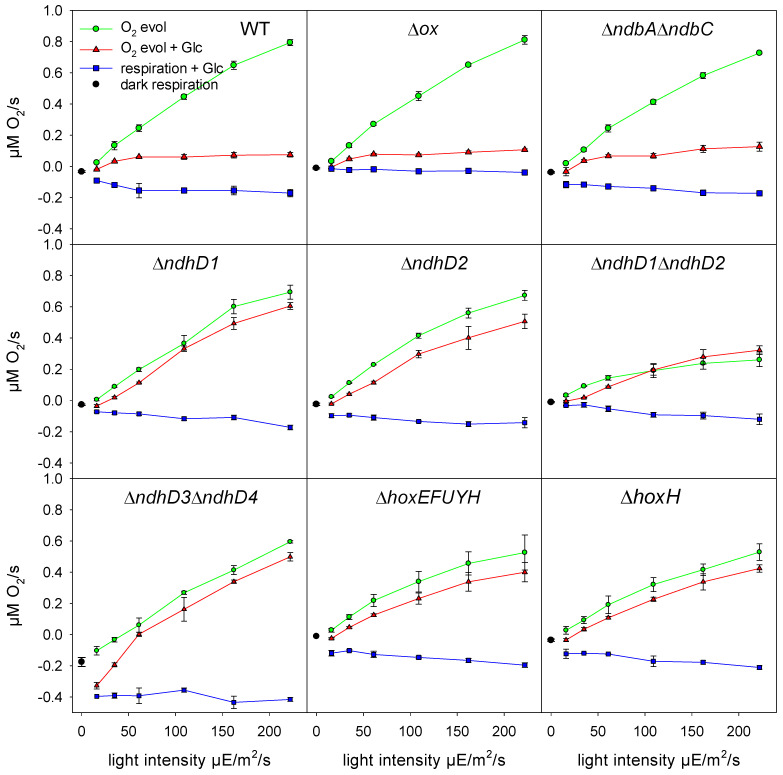
Oxygen evolution without (green lines and circles) and after glucose addition (red lines and triangles) and dark respiration (black circle) and respiration after glucose addition (blue lines and blue squares). The rates are given for a cell density of OD_750_ = 5.7. In all cases at least three different cultures of the same strain were measured and their standard deviation is indicated.

**Figure 6 microorganisms-10-01617-f006:**
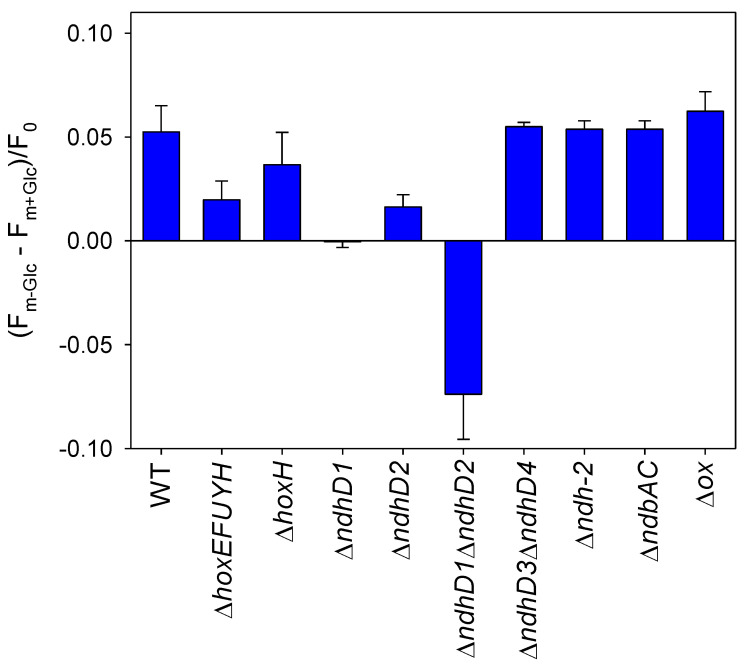
State change after glucose addition as measured by the PAM-fluorimeter of the different deletion strains. The cells were exposed to 120 µE/m^2^/s at the same density as in Figure 4 for about 3 min and the Fm measured (F_m−glc_). After glucose addition the cells were continuously exposed to the same light and after 3 min another Fm was measured (F_m+glc_). The difference between both F_m_ should mirror the shift of antenna away from PSII because of the presence of glucose. At least three different cultures of the same strain were measured. The standard deviation is shown.

**Figure 7 microorganisms-10-01617-f007:**
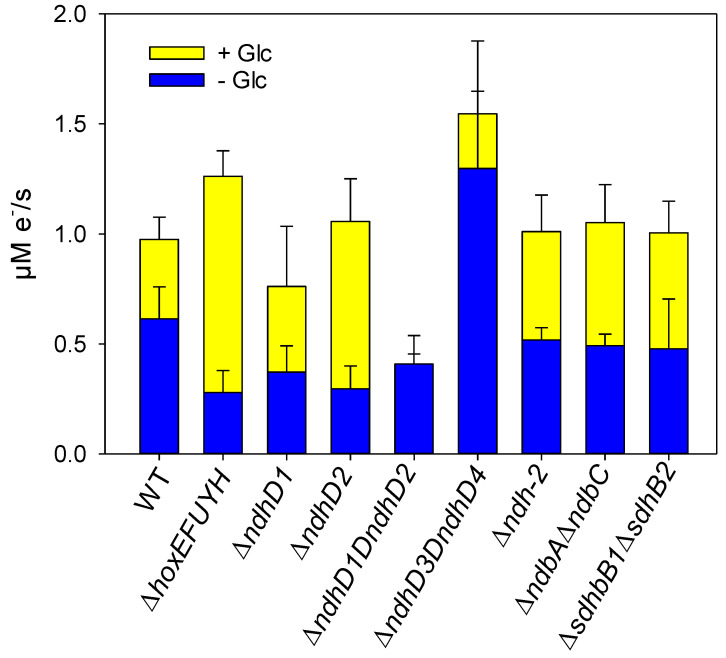
Electron flow with and without glucose in the presence of DCMU for the different deletion strains. Cell suspensions were diluted to OD_750_ = 5.7 and were illuminated with 222 µE/m^2^/s for 3 min before DIRK measurements were started. After the first round of these measurements 10 mM glucose was added and after 1.5 min another set of DIRK measurements was started. The measurements were conducted at least in triplicates. Their standard deviation is indicated.

**Figure 8 microorganisms-10-01617-f008:**
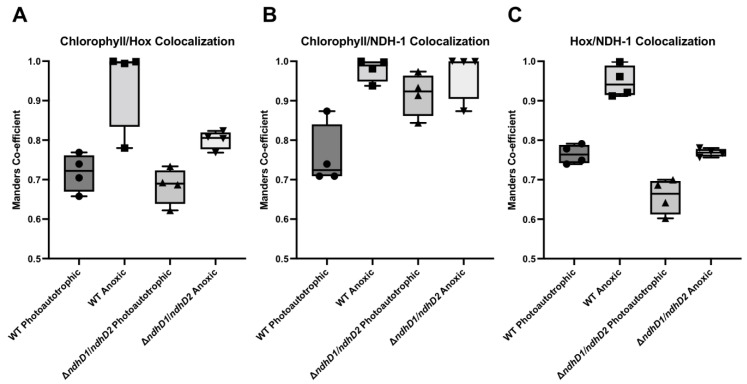
Spatial analysis of Hox, NDH-1 and chlorophyll distribution in cells grown photoautotrophically at 30 µmole photons m^−2^ s^−1^ and following anoxia. The distribution of Hox relative to photosynthetic pigment (**A**), The distribution of NDH-1 relative to photosynthetic pigment (**B**), and the distribution of Hox relative to NDH-1 (**C**). Box plot displays the minimum and maximum Manders coefficient values with individual data points. Data represent four biological replicates: WT photoautotrophic *n* = 898, WT anoxic *n* = 1010, Δ*ndhD1*Δ*ndhD2* photoautotrophic *n* = 912, Δ*ndhD1*Δ*ndhD2* anoxic *n* = 750.

**Figure 9 microorganisms-10-01617-f009:**
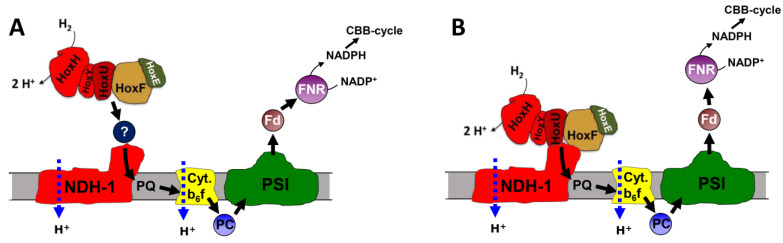
Scheme of the two possible scenarios suggested by our results. (**A**) The bidirectional hydrogenase might either reduce an unknown mediator that needs to have a lower redox potential as ferredoxin 1 or (**B**) it directly binds to the NDH-1 complex and H_2_ oxidation is directly linked to PQ reduction.

**Table 1 microorganisms-10-01617-t001:** ATP/NADPH ratio as calculated from electron flow through PSI (V_max_ + H_2_) and electrons generated by H_2_ oxidation (V_max_H_2_-uptake). If an electron is introduced into the intersystem chain by the NDH-1 complex it should translocate 4 H^+^ into the lumen before passing PSI. In *Synechocystis* 4.66 H^+^ are needed to produce one ATP resulting in the production of 10.6 ATP from 12.4 electrons. On the other hand, when H_2_ is the only electron donor, two electrons from H_2_ oxidation should yield one NADPH. In wild type cells three NADPH are formed from six e^−^ and an ATP/NADPH ratio of 10.6/3 = 3.5 is produced. Standard deviations are given.

Strain	V_max_ + H_2_ e^−^/PSI/s	V_max_ H_2_-Uptake e^−^/PSI/s	ATP/NADPH Ratio ^1^
WT	12.4 ± 0.5	6.0 ± 0.9	3.5 ± 0.5
Δ*ndhD1*	10.2 ± 1.1	5.4 ± 0.9	3.3 ± 0.7
Δ*ndhD2*	10.0 ± 0.4	6.4 ± 0.8	2.7 ± 0.4
Δ*ndhD1*Δ*ndhD2*	9.2 ± 1.6	1.9 ± 0.5	4.0 ± 1.2 ^2^
Δ*ndhD3*Δ*ndhD4*	15.8 ± 1.7	6.4 ± 0.9	4.2 ± 0.8
Δ*ndh-2*	20.6 ± 0.7	14.1 ± 1.4	2.6 ± 0.3

^1^ The ATP/NADPH ratio was calculated from the amount of ATP using V_max+H2_ × 4/4.66 and the amount of NADPH by V_max H2-uptake_/2, assuming that all the e^−^ passing PSI enter the electron transfer via the NDH-1. ^2^ In case of the Δ*ndhD1*Δ*ndhD2* the entry site for electrons into the electron transfer must be predominantly NDH-2 thus only two protons are transferred per electron from the stromal to the luminal site. To calculate the amount of ATP the formula V_max+H2_ × 2/4.66 was used.

## Data Availability

All relevant data are presented in the article. Raw data can be provided on reasonable request.

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
