# Peer review of "Evidence for Electron Transfer from the Bidirectional Hydrogenase to the Photosynthetic Complex I (NDH-1) in the Cyanobacterium Synechocystis sp. PCC 6803"

_microorganisms, 2022, doi:10.3390/microorganisms10081617_

Round 1
Reviewer 1 Report
Overall this is a very strong contribution providing new insight into the function of the NDH1 complex of cyanobacteria. Specifically, it previously hypothesized role of hydrogenase to contribute to the NDH complex based on structural Homeology has found strong experimental support in this work. There are only a few relatively minor aspects that if a dress may strengthen the work and of which, the authors may need to consider:
Is the OD750 a good proxy for cell number--for example are the cells of the different strains have the same scattering properties, size etc.?
The rate units of the y-axis on many of the figures are labeled in micromoles per second. There needs to be greater clarity on the cellular/enzymatic basis of these values--is per cell, Chl etc.
The author should consider the possibility that the CCM associated subunits, NdhD3/4 and NdhF3/4 may compete with the D1, D2 and F1 subunits for occupancy on the core complex and consequently hypothetically the higher rates of respiration and associated features in the mutational absence of the CCM subunits, the higher rates of oxygen consumption etc may be due to higher levels of dedicated respiratory NDH-1 complexes. Similarly, the absence of the CCM subunits and the associated inorganic carbon deficit may cause an up-regulation of the core complexes, perhaps also contributing to higher levels of respiratory NDH-1 complexes. Have the authors attempted to evaluate the accumulation of the various subunits?
Author Response
Overall this is a very strong contribution providing new insight into the function of the NDH1 complex of cyanobacteria. Specifically, it previously hypothesized role of hydrogenase to contribute to the NDH complex based on structural Homeology has found strong experimental support in this work. There are only a few relatively minor aspects that if a dress may strengthen the work and of which, the authors may need to consider:
Is the OD750 a good proxy for cell number--for example are the cells of the different strains have the same scattering properties, size etc.?
We agree that this is an important point to consider. The OD750 is a good proxy for cell dry weight as shown in a previous publication from our group (Makowka et al. 2020). This also applies when comparing wild type and mutant strains. This is now mentioned in section 2.1 of Materials and Methods.
The rate units of the y-axis on many of the figures are labeled in micromoles per second. There needs to be greater clarity on the cellular/enzymatic basis of these values--is per cell, Chl etc.
Since we used OD750 as the reference (see above) this is now indicated in all the figures
The author should consider the possibility that the CCM associated subunits, NdhD3/4 and NdhF3/4 may compete with the D1, D2 and F1 subunits for occupancy on the core complex and consequently hypothetically the higher rates of respiration and associated features in the mutational absence of the CCM subunits, the higher rates of oxygen consumption etc may be due to higher levels of dedicated respiratory NDH-1 complexes. Similarly, the absence of the CCM subunits and the associated inorganic carbon deficit may cause an up-regulation of the core complexes, perhaps also contributing to higher levels of respiratory NDH-1 complexes. Have the authors attempted to evaluate the accumulation of the various subunits?
We did not attempt to detect the different Ndh-subunits. Since this was not in the focus of our work. However, we agree that this is an important aspect that needs to be addressed in future studies. Concerning the role of the different complexes and the higher respiration of the NdhD3D4 mutant we added – as suggested by the reviewer - the following sentences in section 3.1: “We attribute the highly increased respiration in the DndhD3DndhD4 to either a kind of compensatory upregulation of the respiratory complexes NDH-11 and NDH-12 but it might also be caused by a need for higher respiration/ATP production due to the lack of the CCM.“
Reviewer 2 Report
This is an interesting piece of work that provides indirect evidence for an association between the bidirectional hydrogenase and photosynthetic complex I in a cyanobacterium, providing significant new information on the role of the hydrogenase in cyanobacterial electron transport pathways. There is a lack of information in places on replicates and statistics: otherwise I have only fairly minor corrections to suggest. Specific comments (in the order in which they occur in the manuscript):
1. p.3, lines 107-108. I think it should be mentioned here that SDH is known it be a major player in transferring electrons from glucose breakdown to the respiratory electron transport chain (eg ref 32).
2. p.5 lines 214-218. The signals obtained with 633nm excitation and 695-720 nm emission will not solely be from chl a as suggested, because 633 nm excitation predominantly excites the phycobilins, whose fluorescence yield is very high and extends right across the red region of the spectrum. It would be accurate to describe this as fluorescence from the photosynthetic pigments, and I think it should labelled as such. That would mean modifying the labels and legends in Fig 8 and Figs S9 and S10, as well as discussion at various points in the text.
3. Lines 215-217. The text claims that fluorescence spectra were recorded in the microscope, but I do not think this is what is meant. Should be "fluorescence was measured between x and y nm"?
4. lines 218-223. The purpose of the image bleaching is presumably to get a cleaner image of YFP/GFP fluorescence vs background autofluorescence, but this is not explained and no controls are shown to demonstrate that this works. More generally, it would be good to show some controls to demonstrate that it is possible to cleanly distinguish GFP from YFP and both of them from the photosynthetic pigments.
5. Statistics. What are the error bars in Figs 2,4,5,6,7, S2,S3,S4,S5,S8, and what are the error values in Table 1? Standard error or standard deviation?
6. On a related topic, the statistical significance of key differences implied in these figures should be calculated and discussed.
7. Fig. 8. The legend states that the data are from 4 biological replicates, but how many cells were included in the images from each replicate? Also, what is the statistical significance of the differences shown in the Manders coefficients for different conditions?
8. The Supplementary Material carries a different title from the main paper, and one that is not really justified by the data.
Author Response
This is an interesting piece of work that provides indirect evidence for an association between the bidirectional hydrogenase and photosynthetic complex I in a cyanobacterium, providing significant new information on the role of the hydrogenase in cyanobacterial electron transport pathways. There is a lack of information in places on replicates and statistics: otherwise I have only fairly minor corrections to suggest. Specific comments (in the order in which they occur in the manuscript):
- p.3, lines 107-108. I think it should be mentioned here that SDH is known it be a major player in transferring electrons from glucose breakdown to the respiratory electron transport chain (eg ref 32).
We are aware of the experiments done in Ref 32 concerning the respiratory role of the SDH. However, recent investigations reveal that the TCA cycle has a very low flow in Synechocystis (https://academic.oup.com/pcp/article/58/3/537/2959855 ) and behaves more like two independent branches. The oxidizing branch delivers oxoglutarate for N-assimlation while the other branch including the MDH is working more in the reducing direction. See e.g. https://www.sciencedirect.com/science/article/pii/S1096717621000434
In addition, our own results (see Fig. 7) indicate that the respiratory role of the SDH even under the highly oxidizing conditions applied (in the light with DCMU added) must be minor since the sdhB1sdhB2 deletion strain behaves like wild type cells. We consider this subject as unsettled and did not want to get into it.
- p.5 lines 214-218. The signals obtained with 633nm excitation and 695-720 nm emission will not solely be from chl a as suggested, because 633 nm excitation predominantly excites the phycobilins, whose fluorescence yield is very high and extends right across the red region of the spectrum. It would be accurate to describe this as fluorescence from the photosynthetic pigments, and I think it should labelled as such. That would mean modifying the labels and legends in Fig 8 and Figs S9 and S10, as well as discussion at various points in the text.
This comment was addressed by amending text to refer to chlorophyll fluorescence as photosynthetic pigment fluorescence when referencing micrograph images/analysis.
- Lines 215-217. The text claims that fluorescence spectra were recorded in the microscope, but I do not think this is what is meant. Should be "fluorescence was measured between x and y nm"?
This comment was addressed by amending text as suggested; ‘photosynthetic pigments were excited with 633 nm and fluorescence was measured between 695nm-720 nm.
- lines 218-223. The purpose of the image bleaching is presumably to get a cleaner image of YFP/GFP fluorescence vs background autofluorescence, but this is not explained and no controls are shown to demonstrate that this works. More generally, it would be good to show some controls to demonstrate that it is possible to cleanly distinguish GFP from YFP and both of them from the photosynthetic pigments.
This comment was addressed by giving a brief explanation; ‘In order to minimize the autofluorescence signal from the photosynthetic pigments, an initial pre-bleached image was captured followed by image bleaching, briefly samples were exposed to 488 nm and an image was captured after a few seconds. Images were processed using the Fiji software.’ Control figures were also included in the supplementary document (figure S9, Figure S10 & Figure S11).
- Statistics. What are the error bars in Figs 2,4,5,6,7, S2,S3,S4,S5,S8, and what are the error values in Table 1? Standard error or standard deviation?
Many thanks for making us aware of this lack. Now in all the figures and in Table 1 it is indicated that standard deviations are given.
- On a related topic, the statistical significance of key differences implied in these figures should be calculated and discussed.
A pairwise t-test to the WT of e.g. Fig. 2 shows that all are different to the WT control on a p level of 0.001. However, we wanted to be more cautious and discussed only those differences where the standard deviations are outside each others range and considered all others as similar. This is not visible to the reader concerning the data concerning the colocalization of hox and NDH-1. Thus, to enable independent judgement of the statistical significance of this data we provided the p values from a t-test (see next point).
- Fig. 8. The legend states that the data are from 4 biological replicates, but how many cells were included in the images from each replicate? Also, what is the statistical significance of the differences shown in the Manders coefficients for different conditions?
This comment was addressed by the addition of p values from t-Tests between the strains & conditions where appropriate with the cell numbers of each analysis being displayed in figure 8 legend. Methods were amended; ‘with the variance between conditions and strains being measured using T-Test as previously described (Mcdonald & Dunn 2013).’
- The Supplementary Material carries a different title from the main paper, and one that is not really justified by the data.
We are very sorry. The supplementary material gave the title of a previous version in work. It is now correct and the same as the title of the main paper.
Round 2
Reviewer 2 Report
The authors have made a good response to nearly all my comments. My only remaining issue is with my previous comment (4):
**************************
- lines 218-223. The purpose of the image bleaching is presumably to get a cleaner image of YFP/GFP fluorescence vs background autofluorescence, but this is not explained and no controls are shown to demonstrate that this works. More generally, it would be good to show some controls to demonstrate that it is possible to cleanly distinguish GFP from YFP and both of them from the photosynthetic pigments.
This comment was addressed by giving a brief explanation; ‘In order to minimize the autofluorescence signal from the photosynthetic pigments, an initial pre-bleached image was captured followed by image bleaching, briefly samples were exposed to 488 nm and an image was captured after a few seconds. Images were processed using the Fiji software.’ Control figures were also included in the supplementary document (figure S9, Figure S10 & Figure S11).
***********************************
The authors still don't actually explain what they're doing here, so it may be puzzling for some readers. The idea is to selectively bleach out the GFP/YFP fluorescence leaving just the native autofluorescence, then to substract the autofluorescence image from the pre-bleach image to give a cleaner image of GFP/YFP fluorescence. They aren't "minimising autofluorescence" as suggested in lines 231-232, but they are hopefully minimising the contribution of autofluorescence to the processed images. The appropriate control is to do the same for wild-type cells (ie no GFP/YFP tags) and show that the autofluorescence is effectively removed from the processed images. I don't see any such controls in Fig S9 or Fig S10, and there is no Fig. S11. An additional, minor, point there: to avoid confusion, the cells shown in Fig. S9 should not be described as "wild type" because they aren't: this is a GFP/YFP tagged Hox/NDH-1 mutant. What the authors mean is that these are not delta-NDH mutants as shown in Fig S10.
Author Response
The authors still don't actually explain what they're doing here, so it may be puzzling for some readers. The idea is to selectively bleach out the GFP/YFP fluorescence leaving just the native autofluorescence, then to substract the autofluorescence image from the pre-bleach image to give a cleaner image of GFP/YFP fluorescence. They aren't "minimising autofluorescence" as suggested in lines 231-232, but they are hopefully minimising the contribution of autofluorescence to the processed images. The appropriate control is to do the same for wild-type cells (ie no GFP/YFP tags) and show that the autofluorescence is effectively removed from the processed images. I don't see any such controls in Fig S9 or Fig S10, and there is no Fig. S11. An additional, minor, point there: to avoid confusion, the cells shown in Fig. S9 should not be described as "wild type" because they aren't: this is a GFP/YFP tagged Hox/NDH-1 mutant. What the authors mean is that these are not delta-NDH mutants as shown in Fig S10.
We very much appreciate the scrutiny of reviewer 2 and would changed the materials and methods section 2.8 we added the following paragraph (lines 232-244).
Photosynthetic pigments found in both Photosystem I and Photosystem II contain chromophores, which result in background autofluorescence, GFP and YFP both have strong absorption and emission spectrums, which bleach easily under excessive laser illumination at 488 nm and 514 nm [30]. To determine the fluorescence, which directly results from both the GFP and the YFP, images were recorded (pre-bleach), and then the laser intensity was increased for ten seconds to bleach both the GFP and the YFP (post bleach). Post bleach images were then recorded for both the GFP and the YFP. The post bleach images were subtracted from the pre bleached images, providing a true depiction of the distribution of GFP and YFP in the cell (Figures S9 & S10) [30]. Figure S11 depicts a wildtype control (no GFP/YFP), images were captured and then the cells were subject to increased illumination for 10 seconds. There was no impact on the background autofluorescence from the photosynthetic pigments and no contribution from these pigments to the GFP or YFP channels (Figure S11).
We added reference 30 and Figure S11 in the supplementary material to clarify this point.